# Effects of Feed-Through Sulfur on Growth Performance, Atmospheric Ammonia Levels, and Footpad Lesions in Broilers Raised Beginning with Built-Up Litter

**DOI:** 10.3390/ani12172206

**Published:** 2022-08-27

**Authors:** Matthew A. Bailey, Joseph B. Hess, James T. Krehling, Kenneth S. Macklin

**Affiliations:** Department of Poultry Science, Auburn University, Auburn, AL 36849, USA

**Keywords:** poultry, broiler, ammonia, pododermatitis, sulfur

## Abstract

**Simple Summary:**

During commercial farming of chickens, accumulation of feces and moisture in the litter on the floor of the chicken house can lead to ammonia production by natural bacteria in the litter that ingest the chicken feces. Excessive ammonia levels can lead to health problems for the chickens, which can inhibit their growth and cause sores on their breasts and paws. To mitigate this animal welfare issue, this experiment tested the use of sulfur as a feed additive (feed-through sulfur) to acidify chicken feces and prevent ammonia from being formed from bacterial action. In this experiment, feed-through sulfur was tested on three different flocks of chickens. The combination of feed-through sulfur with a traditional litter acidifier, sodium bisulfate, showed improved ammonia control over sodium bisulfate alone in one flock. For the other two flocks, ammonia control was similar between feed-through sulfur with sodium bisulfate and sodium bisulfate alone. In addition, litter acidification was best with the combination of feed-through sulfur and sodium bisulfate in 2 out of 3 flocks. These results indicate the possibility that adding sulfur to feed while also using sodium bisulfate litter treatment may work together to better control ammonia in the chicken house.

**Abstract:**

To the poultry industry, ammonia accumulation within poultry houses can be a costly issue, as this can lead to problems with bird performance, damage to economically important parts such as paws, and customer disapproval due to animal welfare concerns. Common management practices for ammonia control can be quite effective; however, these methods are used variably from farm to farm, which necessitates ammonia control measures that poultry companies can more uniformly implement across all contract growers. One possible measure is ammonia control through feed additives, which would allow poultry companies more direct control over the treatment. This project explored the efficacy of elemental sulfur added directly to the feed (feed-through sulfur) in controlling litter ammonia levels, live performance, and paw quality of broilers raised on built-up litter over three successive flocks. Feed-through sulfur on its own showed inconsistent effects on performance or footpad lesions after 38 days of production compared to sodium bisulfate or control treatments. However, combination of feed-through sulfur and sodium bisulfate showed a potential synergistic effect on ammonia levels and litter pH, although there were few differences between treatments and controls; therefore, additional research must be explored to confirm these observations.

## 1. Introduction

An important issue to the poultry industry is the production and accumulation of ammonia in poultry houses due to microbial metabolization of uric acid excreted by the bird. This is an important ecological issue, and there has been research showing the negative impact of ammonia released from poultry houses on the environment [1]. In addition, ammonia build-up can lead to considerable losses for poultry companies due to health problems in birds associated with excess ammonia. These health problems include burns on the footpad and breasts and irritated or damaged respiratory tissue; excess ammonia ultimately leads to reduced bird performance, condemnation of parts at the processing plant, and potential loss of customers due to animal welfare concerns [2,3,4,5,6,7].

Ammonia production is influenced by factors such as house temperature, moisture content and alkalinity of the litter, and higher ammonium ion concentration [8]. Traditionally, management practices for controlling ammonia include maintaining adequate ventilation in the house, managing litter moisture levels, and the use of acidifying litter amendments such as sodium bisulfate, which helps sequester nitrogen in the litter [1,5]. Other litter additives that have shown efficacy in acidifying litter and reducing ammonia volitalization include propionic acid, superphosphate, phosphoric acid, ferrous sulfate, aluminum chloride, potassium permanganate, and aluminum sulfate [6,9,10,11] Experimentally, biochar has also demonstrated the ability to improve ammonia retention in litter when used as a direct litter amendment [12]. Likewise, humic acids derived from organic compost have also been shown to have a positive effect on litter ammonia levels and intestinal integrity in broilers [13]. While current ammonia management practices can be effective, there can be considerable variation in their implementation across the 100 or more farms that a particular poultry company may contract with. Therefore, it is in the best interest of companies to provide more uniformity in ammonia control methods.

One centralized approach could be to utilize feed additives to reduce ammonia production, as the company provides feed to the grower and would have control over its implementation. There are several published methods showing varied degrees of success with ammonia control in this manner; one method is to use biopreparations of enzymes, including those derived from plant extracts such as *Yucca schidigera*, which can provide various enzymes, saponins, antioxidants, and resveratrol, leading to altered nitrogen metabolism in animals, reduced urea in the blood, and subsequently less ammonia in the air [14,15,16,17]. When used as feed additives, mixtures of aluminosilicates with biochar, and biochar alone have been shown to lower ammonia emissions, although with a negative impact on feed conversion [18]. This effect on ammonia is presumably caused by ammonia adsorption by the aluminosilicates as well as adsorption by carboxyl compounds, phenols, and lactones present on the biochar surface, in addition to changes in the gut microbiome due to biochar supplementation [19,20]. Another approach to ammonia control is to utilize effective microorganisms either as feed additives or additives to the environment, such as bacteria from the nitrogen-fixing genera *Rhizobium*, *Azotobacter, Azospirillum, Beijerinckia, Azoarcus, Burkholderia, Enterobacter, Klebsiella,* and *Bacillus* [21,22,23].

Another method that has shown some effectiveness in lowering ammonia in commercial laying hens is using elemental sulfur as a feed additive, or feed-through sulfur. When included in the diet, feed-through sulfur was able to reduce volatilized ammonia compared to the control by 40–65% depending on the dosage [24]. Previous experiments have shown some efficacy of feed-through sulfur in lowering litter pH and the incidence of footpad lesions in broilers raised on fresh bedding [25]. However, in the industry litter is typically reused for many flocks; therefore, it was necessary to repeat these experiments using more representative methods. To meet this necessity, the following experiments report the effects of feed-through sulfur on bird performance, ammonia levels, footpad lesions, and litter pH in broilers raised on built-up, or previously used, litter.

## 2. Materials and Methods

### 2.1. Live Production Method

Three consecutive flocks of Ross 708 broilers were raised to 38 days of age, and there was a five-day down time between each flock. Experimental facilities consisted of a negative-pressure ventilated, solid-sided house with temperature initially set at 33 °C and gradually reduced to 20 °C at day 21. Photoperiod was 23 L:1D for the first 7 days followed by 20 L:4D for the remainder of the study. Light intensity was 30 lux from 1–7 days, 10 lux from 8–14 days, and 5 lux from day 15 onward. A three-feed diet consisting of starter, grower, and finisher feeds (standard corn-soy formulation [26]) were provided ad libitum in hanging feeders in each pen. In addition, each pen had nipple water lines and previously used, pine shaving-based litter. The litter was generated from research pens in which 3 flocks of birds had been raised to 56 days of age. Floor pens (2.2 m^2^) were utilized for each flock, with 25 birds/pen. For each treatment there were 8 replicate pens, which calculates to 32 pens/flock. Each treatment/pen combination was kept consistent throughout all 3 flocks. Table 1 lists the four treatments, which were applied either directly to the litter (sodium bisulfate) and/or the feed (sulfur). Added sulfur amounts were determined in a preliminary study with laying hens [24]. The treatments consisted of a control with no sulfur or sodium bisulfate (Control), treatment with feed-through sulfur at 2.3 kg/ton of feed (Sulfur), treatment with feed-through sulfur at 2.3 kg/ton and sodium bisulfate at 45 kg/93 m^2^ (Sulfur + SB), and treatment with sodium bisulfate alone at 45 kg/93 m^2^ (SB). On days 9 and 38 for each flock, birds were weighed by pen using IQ Plus 210 digital scales (Rice Lake Weighing Systems, Rice Lake, WI, USA) and feed consumed by pen was weighed to calculate average feed conversion ratio (FCR) and body weight (BW).

### 2.2. Ammonia Measurement

Ammonia was measured on days 9, 23, and 38 for flock 1 and days 0, 9, 23, and 38 for flocks 2 and 3 utilizing a previously published method [27]. On day 0 ammonia was measured before placement of chicks. A plastic box approximately 25 × 17 × 11 cm was connected to a Dräger Chip Measurement System (CMS) (Drägerwerk AG, Lubeck, Germany) by a hose. An ammonia measurement was then taken from each pen by placing the box in the center of the pen, applying an equilibration period of 10 min, and then operating the CMS according to the manufacturer’s instructions. A 1 min purge was performed between each measurement to expel residual ammonia from the hose.

### 2.3. Litter pH Measurement

To measure litter pH for each flock, three 20 g portions were collected on day 38 from each pen, pooled together by pen, and thoroughly mixed to create a composite sample for each pen. The areas sampled included under the water line, next to the feeder, and middle of the pen. From each composite sample, a 5 g portion was removed, thoroughly mixed with 45 mL of distilled water, and allowed to equilibrate for 45 min before taking a pH measurement using a Model 50 electronic pH meter (Fisher Scientific, Waltham, MA, USA) according to the manufacturer’s instructions.

### 2.4. Footpad Lesions

For each bird in all three flocks, footpad lesions were measured visually on day 38 using a previously published three-point scale [28]. The scale was defined as 0 indicating no apparent lesion, a score of 1 indicated a lesion <1 cm, and a score of 2 indicated a lesion >1 cm on the footpad. Lesions were measured by the same person for every bird on the last day of the trial. Scores for individual birds were averaged by treatment (*n* = 200).

### 2.5. Data Analysis

Data was analyzed using the General Linear Model Procedure in SPSS (IBM, Armonk, NY, USA). Any significant results *(p* ≤ 0.05) were then separated using Tukey HSD.

### 2.6. Animal Ethics

These experiments were conducted in accordance with the Institutional Animal Care and Use Committee at Auburn University. The corresponding IACUC PRN number is 2014–2056 

## 3. Results

### 3.1. Live Performance

Table 2 lists the live production data for flocks 1, 2, and 3. For flock 1, on day 38, average FCR data was poorest (*p* = 0.001) for the Sulfur treatment (1.63 kg/kg). The Sulfur + SB treatment (1.59 kg/kg) and the SB treatment (1.58 kg/kg) outperformed Sulfur but were statistically similar to the Control (1.60 kg/kg). No other significant results were detected.

For flock 2, on day 9, average BW values were statistically significant (*p* = 0.022) when comparing the Sulfur treatment (0.168) and Sulfur + SB (0.181). Both Sulfur and Sulfur + SB were intermediate when comparing to SB or Control treatments. On day 38, average BW were statistically significant (*p* = 0.001). BW for the Sulfur treatment (1.92) and Sulfur + SB (1.92) performed worse than the Control (2.11) and SB (2.21) treatments. No other comparisons were significantly different.

For flock 3, day 9 average BW showed significant differences between treatment groups. The Control (0.187) and SB treatment (0.186) were significantly different than Sulfur (0.169). Other than these observations, no other significant results were detected.

### 3.2. Ammonia

Table 3 shows the ammonia data (ppm) for flocks 1, 2, and 3. For flock 1, the ammonia levels were very low even by day 23 for all treatments. At day 9, Sulfur + SB (1.61) and SB treatments (1.64) showed lower ammonia levels than the Control and Sulfur treatments. From an ammonia control perspective, Sulfur performed on par with the Control treatment while Sulfur + SB and SB treatments performed best statistically (*p* = 0.002) by day 9. No other statistical differences were detected between treatments.

For flock 2, day 0 ammonia data was taken at this time as well. What was observed at day 0 was a significant *(p =* 0.001) ammonia reduction for Sulfur + SB (3.41) and SB treatments (3.86) compared to the Control (17.76) and Sulfur treatments (18.18). On day 9, the SB (3.20) treatment was lower than the Control (8.93), but statistically similar to Sulfur and Sulfur + SB, which were not statistically different than the Control. For the Control (8.93) and Sulfur treatments (7.96), ammonia levels dropped by more than half when compared to day 0 values for the same treatments. On day 38, significance (*p* = 0.008) was observed between treatment groups once again.

For flock 3, significance between groups was observed at day 0 and day 9. Treatments containing sodium bisulfate (Sulfur + SB and SB) exhibited significantly lower ammonia levels than non-sodium bisulfate treatments (Control and Sulfur). On day 9, Control and Sulfur treatments were lower than their values on day 0. The Sulfur + SB (1.13) and SB treatments (0.94) were significantly lower than the Control (5.83). Sulfur was intermediate to all three others. No differences occurred for ammonia on days 23 and 28.

### 3.3. Litter pH

Data from litter pH measurements taken on day 38 for flocks 1, 2, and 3 is presented in Table 4. For flock 1, there was no statistical difference in pH between any treatments. However, for flocks 2 and 3, the Sulfur + SB treatment showed lower pH than the Control, whereas there were no other statistically significant differences.

### 3.4. Footpad Lesions

Footpad scores for flocks 1, 2, and 3 are presented in Table 4. There were no significant differences observed between footpad scores for any treatments in all three flocks.

## 4. Discussion

Overall, 2 of the 3 flocks showed no statistical differences between the Sulfur treatments and the SB treatment in terms of performance by day 38 of the study. However, there was no statistical difference between treatments and Control. Therefore, it is difficult to form conclusions about the impact of feed-through sulfur on performance. In addition, observing the day 9 and day 38 average BW data, it appeared the sulfur inclusion could have negatively affected performance in flock 2. For flock 3, it was noted that while there appeared to be a performance concern with Sulfur at day 9, it was not observed by day 38, as all treatments were similar.

As observed in past data [29], sodium bisulfate loses effectiveness on ammonia levels over time. This is reflected in the ammonia measurements for the SB treatment on day 38, especially in flock 2 when ammonia levels for the SB treatment (13.95 ppm) were similar to the Control (12.31 ppm). Although it was day 38 of the trial, the sulfur-containing diets still performed well in terms of ammonia levels. However, it is important to note that this phenomenon was only observed in flock 2. Overall, the Sulfur + SB treatment performed better than SB alone in controlling ammonia in one flock and equivalent to SB in two flocks, although no statistically significant differences were observed between Sulfur + SB and the Control.

Results from litter pH measurements showed lower pH in the Sulfur + SB treatments for two flocks, suggesting that litter quality may be improved by combination of feed-through sulfur with sodium bisulfate when compared to sodium bisulfate alone. It is possible that a synergistic effect between feed-through sulfur and sodium bisulfate allows for continuing pH control even as sodium bisulfate begins to lose effectiveness, as is common 3–4 weeks into production. Although litter quality may have improved with sulfur and sodium bisulfate, this was not reflected in footpad scores, as no significant differences were detected between any treatments for all three flocks. It is also important to note the increased potential for hydrogen sulfide gas production due to added sulfur in the feed. In addition to ammonia, hydrogen sulfide is also harmful to birds at certain concentrations [30,31]. However, hydrogen sulfide levels were not measured in this study; therefore, the authors cannot draw conclusions on the effects of feed through sulfur on hydrogen sulfide levels and bird health.

## 5. Conclusions

The combination of feed-through sulfur with sodium bisulfate-controlled ammonia better in terms of ammonia measurements by day 38 in 1 out of 3 flocks compared to sodium bisulfate alone, and this combination performed similarly to sodium bisulfate alone in 2 out of 3 flocks, indicating that feed-through sulfur may provide synergistic benefits in terms of ammonia control when provided along with sodium bisulfate litter amendment. The sodium bisulfate and feed through sulfur treatment had the lowest litter pH on day 38 in 2 of the 3 flocks. Feed-through sulfur had inconsistent effects on performance or footpad quality over a 38-day production period.

## Figures and Tables

**Table 1 animals-12-02206-t001:** Description of treatments ^1^.

Treatment	Description ^2^
Control	No sulfur added to feed or litter
Sulfur	Sulfur (2.3 kg/t of feed)
Sulfur + SB	Sulfur (2.3 kg/t of feed) and sodium bisulfate (45 kg/93 m^2^)
SB	Sodium bisulfate (45 kg/93 m^2^)

^1^ Sodium bisulfate was added to litter by top dressing the day before placing birds. ^2^ Sulfur treatments were added to the feed and sodium bisulfate treatments were added to the litter.

**Table 2 animals-12-02206-t002:** Live performance of broilers treated with feed-through sulfur and sodium bisulfate litter amendment at days 9 and 38.

Treatment	Days 1–9 Average FCR	Day 9 Average BW (kg)	Days 1–38 Average FCR	Day 38 Average BW (kg)
Flock 1				
Control	1.22	0.232	1.60 ^b^	2.29
Sulfur	1.27	0.231	1.63 ^a^	2.18
Sulfur + SB	1.27	0.229	1.59 ^b^	2.23
SB	1.22	0.232	1.58 ^b^	2.29
Flock 2				
Control	1.38	0.173 ^a,b^	1.60	2.11 ^a^
Sulfur	1.43	0.168 ^b^	1.60	1.92 ^b^
Sulfur + SB	1.36	0.181 ^a^	1.60	1.92 ^b^
SB	1.34	0.180 ^ab^	1.58	2.21 ^a^
Flock 3				
Control	1.37	0.187 ^a^	1.61	2.05
Sulfur	1.40	0.169 ^b^	1.61	2.02
Sulfur + SB	1.34	0.181 ^a,b^	1.60	2.02
SB	1.36	0.186 ^a^	1.61	2.09

^a,b^ Means with different letters were significantly different at *p* ≤ 0.05.

**Table 3 animals-12-02206-t003:** Ammonia levels (ppm) measured on days 0, 9, 23, and 38 in litter from broilers treated with feed-through sulfur and sodium bisulfate litter amendment.

Treatment	Day 0 ^1^	Day 9	Day 23	Day 38
Flock 1				
Control	-	3.08 ^a^	2.42	20.71
Sulfur	-	2.84 ^a^	2.53	16.75
Sulfur + SB	-	1.61 ^b^	2.10	22.11
SB	-	1.64 ^b^	2.22	23.80
Flock 2				
Control	17.76 ^a^	8.93 ^a^	4.02	12.31 ^a,b^
Sulfur	18.18 ^a^	7.96 ^a,b^	2.69	4.89 ^a,b^
Sulfur + SB	3.41 ^b^	4.36 ^a,b^	2.33	3.61 ^b^
SB	3.86 ^b^	3.20 ^b^	2.30	13.95 ^a^
Flock 3				
Control	15.14 ^a^	5.83 ^a^	2.20	12.81
Sulfur	10.73 ^a^	4.90 ^a,b^	1.91	9.53
Sulfur + SB	1.09 ^b^	1.13 ^b^	1.40	4.56
SB	0.86 ^b^	0.94 ^b^	1.21	6.21

^1^ Ammonia measurements were not taken on Day 0 for Flock 1; ^a,b^ Means with different letters were significantly different at *p* ≤ 0.05.

**Table 4 animals-12-02206-t004:** Litter pH and average footpad score at day 38 in broilers treated with feed-through sulfur and sodium bisulfate litter amendment.

Treatment	pH	Average Footpad Score
Flock 1		
Control	7.26	0.016
Sulfur	6.96	0.032
Sulfur + SB	7.3	0.024
SB	7.16	0.047
Flock 2		
Control	7.48 ^a^	0.05
Sulfur	7.09 ^a,b^	0.08
Sulfur + SB	6.95 ^b^	0.04
SB	7.31 ^a,b^	0.03
Flock 3		
Control	7.83 ^a^	0.011
Sulfur	7.58 ^a,b^	0.016
Sulfur + SB	7.13 ^b^	0.049
SB	7.46 ^a,b^	0.017

^a,b^ Means with different letters were significantly different at *p* ≤ 0.05.

## Data Availability

The data presented in this study are available on request from the corresponding author.

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
