# Peer review of "Effects of Feed-Through Sulfur on Growth Performance, Atmospheric Ammonia Levels, and Footpad Lesions in Broilers Raised Beginning with Built-Up Litter"

_animals, 2022, doi:10.3390/ani12172206_

Round 1

Reviewer 1 Report

The aim of the research was to determine the effect of feed-through sulfer on growth performance, atmospheric ammonia levels, and footpad lesions in broiler chickens raised beginning with built-up litter. The number broilers used in the experiment is sufficient (n = 200 for treatment). The Materials and Methods chapter requires corrections and additions. The applied research methods are correct. The discussion is well conducted and comprehensive. Well-chosen references, but the number of references is too small. Before publishing in Animals, the article requires additions and corrections. The proposed changes are listed below:

General comments:

Please prepare the article in accordance with the instructions for authors.

·         For affiliates, the first name and surname initials for each co-author of the article should be provided, the same as given in the "Author Contributions" chapter. For example, J.B.H., K.S.M and e-mail for each co-author

·         Please complete the "Citation" details on the front page

·         When describing the significant, use lowercase "p" in italics, spaces before and after "<" for example (p < 0.05)

·         Please use the abbreviated name journal for item number 2, 3, 11, 12 in References chapter

·        There must be "dot" after every parts abbrevaited name journal, for example, "Poult. Sci." instead of „Poult. Sci”

·        The number should be followed by the units of measurement for the examined features

Detailed comments:

L2 I propose "growth performance" instead of „bird live performance”

L30 little effect? Is this the case see BW 9 d, FCR 1-38 days

L56 + add information about deodorants (based on Yucca Shidiger), preparations of effective microorganisms (EM), aluminosilicates, humic raw materials (peat, charcoal). calcium diphosphate (superphosphate), etc. reducing NH3 emissions, and ways to regulate dietary composition to reduce nitrogen emissions. This will allow you to increase your References to the required 30 items.

L71 Add information about building type (closed and without windows)?, temperature, humidity, photoperiod (length, color, intensity, type). Are there maximum permitted levels in the building for poultry for NH3, CO2 as in the EU (20 ppm and 3000 ppm, respectively) in USA.

L84 what scales was used to determine FI and BW

L124 delete "statistically"

Table 2, FCR only Day 38 not Days 1-38 ?,

L236 + add Figure with "Mortality" days 1-38 if you have information

L124 + add units of measure for numbers

In table 3 - 2.10. 23.80, 3.61b - instead of current form

When was the NH3 concentration measured on Day 0?

L155 p < 0.05 ??

L184, 184 add "ppm"

L201 better - in terms of what?

L206 but "litter" positive or negative - there are statistically significant differences

Author Response

Thank you for taking the time to review the manuscript. Attached are our responses.

Reviewer 2 Report

This manuscript is novel and the experiments were done properly. Authors provide novel findings and they managed to better control ammonia in the chicken house by the combination of feed-through sulfur and sodium bisulfate. However, authors need a minor revision here before acceptance.

  1. Line 19: “adding feed to sulfur” or “adding sulfur to feed”?

  2. Why did the authors not detect the feed intake of broilers in each group?

  3. Line 14: In addition to ammonia, hydrogen sulfide is also a common harmful gas in chicken houses. It is suggested that the authors refer to some literatures for a brief introduction to this part, such as doi: 10.1016/j.ecoenv.2021.112488 and doi: 10.1016/j.jhazmat.2020.124682.

  4. Line 47: The references are old, and the authors are suggested to replace them, such as doi: 10.1016/j.jhazmat.2020.122605, etc.

  5. Line 73: What is the difference between starter, grower and finisher? It is suggested that the authors list the detailed formula of each feed in detail.

  6. Line 81-83: What is the basis for the authors to determine the dosage of sulfur and SB?

  7. Line 85: “FCR and BW”, the full name shall be indicated for the first time.

  8. Line 100: Why did the authors mix the litters of the individual pens together? In this case, it seems that there is no longer a sample parallel and is such data statistically significant after comparison?

  9. Line 112-114: The authors should indicate the sample size here.

  10. Ammonia and hydrogen sulfide are the most harmful gases to chickens in chicken farms. Although the ammonia concentration in the chicken house is reduced through the combination of sulfur and SB. However, will the hydrogen sulfide concentration in the chicken house be increased due to the increase of sulfur? The authors need to explain this problem in the discussion.

Author Response

(The authors gave the same response as above.)
